# An Overview of Reactive Oxygen Species Damage Occurring during In Vitro Bovine Oocyte and Embryo Development and the Efficacy of Antioxidant Use to Limit These Adverse Effects

**DOI:** 10.3390/ani14020330

**Published:** 2024-01-21

**Authors:** Jessica A. Keane, Alan D. Ealy

**Affiliations:** School of Animal Sciences, Virginia Polytechnic Institute and State University, Blacksburg, VA 24060, USA; jakeane@vt.edu

**Keywords:** antioxidants, bovine, in vitro production (IVP), oocyte, embryo, cysteine, cysteamine, selenium, zinc, hypotaurine

## Abstract

**Simple Summary:**

The in vitro production (IVP) of bovine embryos has increased in popularity in the past few decades with improvements in our ability to harvest oocytes from genetically elite heifers and cows, fertilize them in vitro, and successfully culture embryos to the stage where they can be transferred. One issue with IVP is that the efficiency of embryo production is less than ideal. One likely reason for the poor development of IVP embryos is the excessive exposure of oocytes and embryos to reactive oxygen species (ROS). These molecules are produced as a normal by-product of energy generation; however, excess accumulation of ROS during the IVP of oocyte and embryo culture will damage nucleic acids, lipids, and proteins in ways that compromise development potential and post-transfer embryo survival. Molecules that react with and block ROS-induced damage are commonly referred to as antioxidants. This review explores the use of five common antioxidants to limit ROS-induced damage and promote the IVP of embryos.

**Abstract:**

The in vitro production (IVP) of bovine embryos has gained popularity worldwide and in recent years and its use for producing embryos from genetically elite heifers and cows has surpassed the use of conventional superovulation-based embryo production schemes. There are, however, several issues with the IVP of embryos that remain unresolved. One limitation of special concern is the low efficiency of the IVP of embryos. Exposure to reactive oxygen species (ROS) is one reason why the production of embryos with IVP is diminished. These highly reactive molecules are generated in small amounts through normal cellular metabolism, but their abundances increase in embryo culture because of oocyte and embryo exposure to temperature fluctuations, light exposure, pH changes, atmospheric oxygen tension, suboptimal culture media formulations, and cryopreservation. When uncontrolled, ROS produce detrimental effects on the structure and function of genomic and mitochondrial DNA, alter DNA methylation, increase lipid membrane damage, and modify protein activity. Several intrinsic enzymatic pathways control ROS abundance and damage, and antioxidants react with and reduce the reactive potential of ROS. This review will focus on exploring the efficiency of supplementing several of these antioxidant molecules on oocyte maturation, sperm viability, fertilization, and embryo culture.

## 1. Introduction

The in vitro production (IVP) of bovine embryos has gained popularity over the past several decades in improving the accessibility and trade of valuable dairy and beef genetics. Over 1.5 million IVP bovine embryos were generated in 2021 [1]. The popularity of the IVP of embryos is occurring despite lingering concerns with the inefficiency of IVP, reduced post-transfer pregnancy rates, increased pregnancy loss, and the occasional elevated incidences of chromosomal abnormalities and large offspring syndrome [2,3,4,5]. These problems likely exist, at least in part, because we are not able to adequately mimic the oocyte maturation, fertilization, and embryo development processes that occur within the follicle, oviduct, and uterus.

Substantial research has investigated optimal media formulations for culturing embryos [6,7,8,9,10], but deficiencies in the system remain. We contend that additional culture supplements must also be considered to improve the developmental potential of the oocyte and embryo. One class of culture supplements that have been examined are antioxidants. These molecules manage reactive oxygen species. Reactive oxygen species (ROS) are normal by-products from oxygen metabolism produced by the electron transport chain. Cellular stress and/or imbalances of biological chemicals designed to prevent ROS-induced damage can damage DNA, proteins, and lipid bilayers in ways that compromise oocyte quality and maturation potential and reduce embryo development. The adverse effects of ROS are especially relevant during IVP because the repeated exposures to atmospheric oxygen concentrations, light, temperature, and pH alterations are noted sources of ROS production. This review will describe how ROS and other reactive molecules are produced, primarily during IVP, and illustrate how these reactive molecules are normally controlled within cells. We will also explore the efficacy of supplementing antioxidants to improve the efficiency of the IVP of embryos.

## 2. Formation of Reactive Species

The term “reactive species” refers to oxygen (O_2_) and nitrogen molecules that contain one or more unpaired electron(s). The term “free radical” refers to molecules that are highly reactive due to the unbound electron(s) attempting to regain electron stability by oxidizing with other molecules [11]. Nearly all biomolecules can react with free radicals, but nucleic acids, lipids, and proteins are the macromolecules most commonly oxidized by free radicals [12]. Another less commonly discussed category of reactive species are nonradicals. These molecules are less reactive than free radicals, but they remain important to discuss because they often serve as intermediates in oxidation and/or reduction reactions that generate and/or inactivate free radicals.

### 2.1. Reactive Oxygen Species

Numerous free and nonradical O_2_ species exist (Figure 1A), but those that are more commonly associated with cell damage include superoxide anion (O_2_^•−^), hydroxyl radical (^•^OH), hydroperoxyl, peroxyl, and alkoxyl radicals [11,13]. Oxygen nonradicals include hydrogen peroxide (H_2_O_2_), hypobromous acid, hypochlorous acid, ozone, singlet oxygen, organic peroxides, nitric oxide (NO^•^), peroxynitrite (ONOO^−^), and peroxynitrous acid [11,13]. Each of these ROS can be generated within various cellular compartments, but mitochondria are the main source of their production [14]. Mitochondria are necessary for the generation of adenosine triphosphate (ATP) in aerobic cells via oxidative phosphorylation events mediated by electron transport. As ATP is being generated, there is a constant low rate of electron escape from the proton gradient created with electron transport [15], and these electrons will quickly react with O_2_ to generate O_2_^•−^. An enzyme, termed superoxide dismutase (SOD), localized primarily in the matrix or inner membrane of the mitochondria, will convert this ROS to H_2_O_2_ [16]. This nonradical can move across the mitochondrial membrane through specific aquaporin channels called peroxiporins [17,18]. There are two mitigation pathways to reduce H_2_O_2_ into O_2_ and water (H_2_O). The first is the catalase (CAT) enzyme system. The catalase enzyme converts two H_2_O_2_ to generate O_2_ and H_2_O. The second is the glutathione (GSH) peroxidase (GPx) enzyme system, where GPx utilizes GSH as an electron donor to convert H_2_O_2_ to H_2_O. These intrinsic mitigation pathways will be discussed in depth later in this review. Extensive reviews of the chemical formation of ROS and production from oxidative phosphorylation exist [11,14,15,19].

The failure of these ROS enzyme systems to quickly react with these ROS molecules will enable them to generate other, more detrimental ROS (Figure 1A). A prime example is allowing O_2_^•−^ to react with metal ions and iron–sulfur clusters that, in the presence of H_2_O_2_, will generate ^•^OH [20]. Another example is when H_2_O_2_ reacts with chlorine ion (Cl^−^) to generate the highly reactive hypochlorite molecule or with transition metals, such as Cu or Fe, in Fenton and Haber–Weiss reactions to generate the highly reactive ^•^OH [21,22]. Damage caused by ^•^OH has received lots of attention because it rapidly damages nucleic acids, proteins, lipids, and carbohydrates. This damage is especially concerning because, to date, there are no known enzymatic processes identified to inactivate ^•^OH.

While mitochondria are considered the main producers of cellular ROS, the endoplasmic reticulum (ER) is also an important organelle to consider when discussing ROS (Figure 1A). The ER handles protein structural modifications, including disulfide bond formation. Stressed ER will generate H_2_O_2_ from excessive oxidation that occurs from enzymes that control disulfide bond formation. There are also detrimental effects that ROS have on the primary and secondary structures of proteins [23,24,25]. These will be discussed later in this review.

### 2.2. Nitrogen Species

There are two reactive nitrogen species produced in the cytoplasm to highlight: NO^•^ and ONOO^−^ [26] (Figure 1A). Nitric oxide is a signaling molecule used for controlling vascular tone as well as for neurotransmission, immunity, and other cell signals [27]. Its production occurs from the metabolism of L-arginine into L-citrulline and NO^•^ through a family of nitric oxide synthase enzymes localized within different tissues of the body [14,28]. In addition, NO^•^ will generate the highly reactive radical, ONOO^−^, by reacting with O_2_^•−^. Peroxynitrite reacts quickly with several biological molecules, including electron transport chain complexes, which leads to their inactivation [29]. This and other reactive nitrogen species are as damaging as ROS, but less is known about their actions within the oocyte and embryo. Therefore, we will be focusing namely on ROS-induced damage from this point forward.

### 2.3. Detrimental Effects

As mentioned previously, the three main targets of ROS-induced damage are nucleic acids, lipids, and proteins (Figure 1B). The primary type of damage is DNA lesions, where alterations in the base structure occur [30]. This results in single-strand breaks (SSB) or, the more severe, double-strand breaks. The most common DNA modification from oxidative damage is the ^•^OH reaction with deoxyguanosine at the C8 position to form 8-oxodeoxyguanosine (8-oxo-dG) [31]. Hydroxyl radicals also react with imidazole ring-opened products and specifically the C5-C6 double bond of pyrimidines and C8-C5 bonds of purines [32]. The repair of ROS-induced damage is controlled by various base excision repair, nucleotide excision repair, and nucleotide incision repair pathways [32,33]. Mitochondrial DNA (mtDNA) is keenly susceptible to oxidative damage given its close proximity to ROS production and the lack of histones, which play a protective role in ROS-induced damage [34]. Oocytes are particularly susceptible as they contain over 100,000 mitochondria compared to other cells within the body that contain 1000–2500 mitochondria [35].

The DNA methylome is another target for ROS-induced damage. An overall hypomethylation phenomenon occurs with ROS-induced damage, primarily because the pre-described ROS-mediated creation of 8oxy-dG will prevent cytosines within CpG islands from being methylated [36,37]. The indirect route of ROS-induced damage occurs because ROS target S-adenosyl methionine (SAM), a commonly used methyl donor, thus reducing the methylation potential. Other indirect actions of ROS include targeting the enzymes that control demethylation and remethylation DNA methyltransferases (DNMTs) and ten-eleven translocation (TETs) [37]. These aspects are extremely important during early embryogenesis as the mammalian embryo undergoes massive epigenetic reprogramming prior to implantation. Demethylation is rapid in the paternal chromatin (before first cleavage) and slower in the maternal chromatin (first few cleavages), and then a de novo re-methylation process must take place before the embryo forms a blastocyst [38]. Blastocyst formation also requires reconstruction of the methylation landscape within the inner cell mass (ICM) and trophectoderm (TE) [38]. These methylation events are sensitive to external environmental disturbances, including ROS-induced damage.

Another major target molecule for ROS are the lipid bilayers of the cell, mitochondria, nucleus, and other organelles. The primary ROS involved with this damage is ^•^OH, where a peroxidation reaction occurs within lipid methylene groups [39]. This lipid peroxidation compromises membrane fluidity, damages its functional integrity, and limits the ability of membrane-bound proteins (e.g., receptors, ion channels) to function properly. Another source of lipid-targeted ROS-induced damage occurs when various ROS react with O_2_. This generates a peroxyl radical, which will react with polyunsaturated fatty acids to form hydroperoxide (HOO^−^) and alkyl radicals [40]. This type of damage is very concerning because the resulting production of HOO^−^ and alkyl radicals causes a feed-forward reaction, where these radicals will react with nearby fatty acids to generate more radicals [41]. Additionally, excessive production of ROS will compromise mitochondrial permeability transition (mPT) pores. This alters the conductance state of the mitochondria. Low conductance is reversible, but high conductance produces an accumulation of calcium that causes irreversible transmembrane potential that leads to apoptosis due to the release of cytochrome-c that activates the caspase pathway [42].

Protein damage by ROS occurs in several ways. Protein oxidation begins with the reaction of ^•^OH to generate an alkyl radical. The alkyl radical reacts with O_2_ to generate a peroxide radical. Further reaction with an adjacent protein forms a HOO^−^ and alkyl radical that can further generate an alkoxy radical [40]. Oxidative stress and production of ^•^OH can cause protein modification and loss of conformation through the oxidation of amino acid side chains, protein cross linkage, and protein fragmentation [40,43,44]. Conformational changes in proteins can lead to aggregation, fragmentation, distortion of secondary and tertiary structure, and diminution of normal function [44]. Loss of enzymatic activity can happen as a result of conformational change [40]. These factors contribute to the formation of disease states (Figure 1B).

### 2.4. Accumulation of Reactive Species with In Vitro Production

Oxidative phosphorylation is the primary source of energy used in embryo development between the 1-cell and morula stages [45]. Embryos that are more metabolically active early in development generate more ROS [46]. Although these embryos may develop at a similar or even greater rate than other embryos, they often may not have the ability to generate pregnancies at the same efficiency as embryos that will develop properly with less energy demands. These embryos with reduced metabolism are coined as “quiet” embryos [46]. A presumptive reason why these “quiet embryos” contain greater competency for generating pregnancy is because they have undergone less oxidative damage, thanks to their low metabolic rate [30]. In recent years, the quiet embryo hypothesis has been re-evaluated and is now more often referred to as the “Goldilocks Hypothesis” because the “ideal” embryos are those that do not require too much or too little energy during the early stages of development [47]. Another intriguing feature of IVP systems is that most IVP embryos have greater glycolytic activity than embryos developing in utero [48]. This is from increased exposure to atmospheric O_2_ that increases consumption of pyruvate and subsequently increases ROS production [49]. Overall, the metabolic activity is used to indicate pregnancy success from IVP embryo [50].

Various external environmental factors contribute to the excess accumulation of ROS during IVP, and these factors undoubtedly contribute to the suboptimal development of these embryos. Some of the best studied factors include exposure to atmospheric O_2_ conditions, ultraviolet (UV) light, and alterations in temperature and pH. Oxygen conditions vary between 2 and 9% within the reproductive tract in mammalian species [51]. One of the major hurdles in overcoming limitations with the IVP of embryos occurred by culturing embryos in 5–6% O_2_ rather than in atmospheric O_2_ (21%). A summary of findings from four studies where embryos were placed in 5% or 20% O_2_ conditions (Figure 2) illustrates just how profound the improvement in the IVP of bovine blastocyst development can be when embryos are maintained in low O_2_ conditions [7,52,53,54,55].

Most IVP protocols employ atmospheric O_2_ during in vitro maturation (IVM) and in vitro fertilization (IVF) as low O_2_ conditions will generally reduce embryo production efficiency [56,57,58]. However, there is evidence suggesting that lower O_2_ tension is at least as proficient and, in some instances, may be preferred to culturing oocytes in high O_2_. At least two reports have found that oocyte competency can be restored in oocytes cultured in low O_2_ by modifying glucose concentrations in the maturation medium [56,57]. A recent abstract also indicated that reducing O_2_ concentrations during IVM and IVF improves the cryosurvivability of embryos produced with IVP [59]. These findings indicate that further investigation into the merits of using low O_2_ conditions during IVM and IVF may be warranted, especially if we consider endpoints other than embryo development.

Embryos are also suspectable to light damage. Fluorescent lighting contains UV rays, and as little as 3 min of exposure to UV light decreased development of hamster embryos [60]. There is a direct link between UV radiation and DNA damage, where photosensitizers will stimulate ROS production and lead to oxidative DNA damage [61]. When light strength was reduced, blastocyst yield was increased compared to greater intensities and lower ROS production occurred [62,63]. Additionally, altered light exposure may reduce the stress caused by light [64].

Adequate pH balance and proper temperature are key for optimizing IVP success. Media utilized for IVP have been equilibrated under CO_2_ gas to maintain a pH to support embryonic growth. In mammalian cells, intracellular pH is regulated by HCO_3_^−^/Cl^−^ exchangers and Na^+^/H^+^ exchangers [65]. This is especially important in the oocyte and early developing embryo. One study determined that denuded oocytes lack the ability to maintain an internal pH of 7.1 compared to blastocysts [66]. Therefore, removal of IVP plates to manipulate, apply treatment, or undergo a necessary procedure (e.g., fertilization) for oocytes and embryos can influence pH and embryo developmental potential. Covering media with mineral oil not only limits media evaporation but it also can assist with maintaining the pH [67,68]. Without the utilization of mineral oil, the pH can increase within the first 1–2 min of plates being exposed to a non-gassed atmosphere and take 30–35 min to re-equilibrate [69]. Using mineral oil increases the allotted time before media begin to increase in pH level, where small increases are seen to start after 10 min of atmospheric exposure when oil is not used [67,70]. Another important consideration to keep in mind is that ROS production within embryos can alter the pH gradient within the mitochondria as changes in electron transport and, ultimately, the establishment of the H^+^ gradient will be altered when ROS production is elevated [71,72]. Temperatures for IVP culture systems occur close to rectal temperatures in cattle (37–39 °C), even though cultures at lower temperatures do impact embryonic development [73]. However, fluctuations in culture temperatures contribute to detrimental embryonic development [74]. During IVM, mouse and human oocytes are more susceptible to temperature fluctuations that can result in disruption of meiotic spindle assembly [75,76]. An important note to include is that temperature fluctuations may alter metabolic activity. In one study, bovine blastocysts cultured at lower temperatures (37 °C) had reduced amino acid consumption and production [77]. Additionally, GPx activity was reduced in lower culture temperatures (36.5 °C vs. 38.5 °C) [78]. Therefore, it can be deduced that altered/elevated temperatures subsequently increase ROS production (e.g., alterations in mitochondria) and that stable/lowered temperatures reduce ROS accumulation as metabolic demands are lowered. Regardless, both pH and temperature are external environmental factors that can be controlled to improve IVP.

Cryopreservation is another source for ROS accumulation within embryos. This procedure clearly has substantial benefits, including preservation of genetic material, ease with transport around the world, and as a convenience for completing ET. However, cryopreserved bovine embryos are less successful at generating a pregnancy than non-frozen embryos produced by IVP [4]. The oxidative stresses produced include the accumulation of H_2_O_2_, NO, and O_2_^•−^ [79]. As discussed previously, these and other ROS are linked to proteomic, epigenetic, transcriptomic, and genomic changes in bovine embryos [80,81]. On a related topic, ROS accumulation also occurs in cryopreserved semen and subsequent damage caused by ROS includes alterations in calcium fluxes and membrane fluidity, integrity, and permeability [82].

## 3. Intrinsic ROS Mitigation Systems

There are three key enzymes that reduce ROS accumulation within cells (Figure 3A). The first is SOD. It reduces O_2_^•−^ to O_2_ and H_2_O_2_. Three isoforms exist: SOD1 (Cu/Zn-SOD; cytoplasm and nucleus localization), SOD2 (Mn-SOD; mitochondria localization), and SOD3 (EC-Cu/Zn-SOD, extracellular) [83]. Copper (Cu^2+^) is used as cofactor for SOD1 and SOD3 in a 2-step process where Cu^2+^ is reduced to Cu^1+^ to oxidize an O_2_^•−^ molecule to O_2_, and then Cu^1+^ is oxidized to Cu^2+^ by reducing a second O_2_^•−^ molecule into H_2_O_2_ [84]. Iron is used as a cofactor by SOD2 to achieve the same result [85]. The second ROS mitigating enzyme system is the catalase (CAT) system. Its function is to further reduce H_2_O_2_ generated by SOD and convert it to H_2_O and O_2_ [12]. The third set of ROS mitigating enzymes are those enzymes that utilize GSH. This is tripeptide thiol antioxidant that is generated by most cells through two ATP-dependent reactions: one that involves the formation of glutamate cyclase and a second that adds glycine and γ-glutamylcysteine. The formation of GSH is controlled by substrate availability, with cysteine availability usually being the rate-limiting substrate [86,87]. The renewal of GSH occurs through the γ-glutamyl cycle. This process works through enzymatic control by glutathione peroxidase (GPx) that oxidizes GSH to remove H_2_O_2_ and generate H_2_O molecules. The oxidized form of GSH, GSH disulfide (GSSG), is enzymatically reduced by GSH reductase to GSH [88]. Eight GPx family members exist in mammals [15].

These GSH-based enzyme systems are essential for supporting oocyte and early embryonic development. Stores for GSH begin to rise during germinal vesicle breakdown, peak during metaphase II, and decline during zygote formation. Cytoplasmic GSH is at its lowest during the 2 to 8-cell stages and increases thereafter [89,90]. Elevated stores of GSH are necessary during maturation, fertilization, and early development as they are essential for forming, maintaining, and protecting meiotic spindles, reducing the disulfide bonds of the male pronucleus after fertilization, and supporting development past the 2-cell stage [91,92,93,94,95]. As mentioned, de novo synthesis of GSH can occur in most cells. The oocyte accumulates GSH until it matures to the MII phase. Afterwards, no new GSH is produced until after implantation in the mouse [96]. Transport of GSH components and extracellular GSH is facilitated by the transport from blood plasma to the follicles and uptake by cumulus cells and transport through gap junctions to the oocyte. Removal of cumulus cells debilitates the oocytes’ capabilities to synthesize GSH [97,98].

## 4. Antioxidants Examined in Bovine Embryos

The first calf produced by IVP was born in 1981, and since then a substantial amount of emphasis has been placed on improving efficiency rates and success in developing embryos produced by IVP. Many putative, as well as well-established, antioxidants have been characterized for their ability to improve bovine IVP systems. We have identified five antioxidants that have been heavily highlighted for their antioxidant potential in bovine IVP systems (Figure 3B). The section will provide a brief overview of the biological and antioxidant activities of each molecule and a final section will present findings from using these antioxidants during bovine IVP. Several additional antioxidants could have very easily been included in the following discussion. Melatonin and Resveratrol are excellent examples of other antioxidants studied extensively during IVM. Reviews of these molecules already exist [99,100,101,102], so we decided to focus our discussions away from these antioxidants.

### 4.1. Cysteine

Cysteine is the rate-limiting amino acid in GSH production. It is an unstable molecule outside of the cell which undergoes auto-oxidation to form cystine [103]. However, cystine can be reduced back to cysteine within the cell by reacting with β-mercaptoethanol or cysteamine [87,104]. One common way to provide supplemental cysteine is by supplementing N-acetyl cysteine (NAC). Within the cell, NAC is deacetylated to from cysteine. A direct action of NAC also exists, and the presence of a thiol group permits NAC to serve as an electron donor for reactions with ^•^OH, H_2_O_2_, and other ROS [105]. Transport of NAC into the cell occurs without the facilitation of a carrier protein [105,106]. It is important to note that NAC is unable to maintain sufficient levels of cysteine to maintain adequate stores of GSH.

### 4.2. Cysteamine

Cysteamine is another component of GSH, so it is not surprising that, like cysteine, it has been explored as an antioxidant in bovine oocytes and embryos. One of its main functions is to facilitate cysteine availability in two ways: by reducing cystine to cysteine within cells and promoting cysteine uptake by cells [90,107,108].

### 4.3. Selenium

Selenium (Se) is a trace mineral that acts as a cofactor for several enzymes. Its consideration as a molecule with antioxidant potential is proposed because it is a cofactor for GPx1/2/3/4 [109]. As previously discussed, GPx regulates H_2_O_2_ concentrations [110,111]. Selenium deficiencies are associated with infertility in humans [112,113]. Consumption of Se, when not provided in a supplement, is dependent on soil content and absorption into crops and animals [112]. If not provided, Se deficiencies can alter hormonal profiles and decrease pregnancy success rates [114]. Selenium is provided in the form of an inorganic compound (e.g., sodium selenite) or as organic compounds (Se yeast, selenomethionine, selenocysteine) [115,116,117]. Organic Se has easier absorption and higher bioavailability, while inorganic Se is more cost effective [118]. For both organic and inorganic Se, the common intermediate is selenide. Inorganic selenite is easily reduced to selenide by red blood cells. Selenide produced from red blood cells will be excreted into the bloodstream and bound to albumin for further processing within the liver [117]. Unlike selenite, inorganic selenate is not readily reduced to selenide. Instead, selenate is transferred directly to the liver. Both selenide that is derived from inorganic selenite and selenate are utilized by the liver for the synthesis of selenoproteins and cellular GPx [117].

### 4.4. Hypotaurine

Hypotaurine (HTU) functions as an antioxidant outside of the scope of SOD, CAT, and GSH. It is a nonessential amino acid generated from cysteine degradation and pantothenate synthesis [119,120]. Hypotaurine is the precursor for taurine production, and this dehydrogenase reaction requires H_2_O_2_ as a substrate [119]. This HTU to taurine reaction will occurs by using O^2•−^ and hydrogen ions as substrates, and the peroxytaurine intermediate is rapidly converted to taurine and H_2_O_2_. Hypotaurine also acts through a nonenzymatic pathway to directly react and inactivate ^•^OH [121]. The presence of HTU synthesis in gametes and embryos and its presence within the reproductive tract are primary reasons as to why it has been tested for is antioxidant activity in bovine IVP systems [122].

### 4.5. Zinc

Zinc (Zn) is an essential trace mineral involved in numerous cellular functions, including as an enzyme cofactor for DNA methylation, DNA repair, and apoptosis [109,123]. Chronic deprivation of Zn leads to increased oxidative-mediated cell damage [124]. To offset dietary deficiencies, Zn has become a common feed ingredient without inducing Zn toxicity and is provided as organic or inorganic supplements [125]. Bulls supplemented with Zn had improved sperm motility compared to non-supplemented males [126]. Adequate Zn in diets is needed in cattle to limit placental retention after calving [127,128]. As mentioned, Zn is a main cofactor for SOD1 and SOD3. It does not act as an electron donor but rather is required to provide the proper tertiary structure for SOD1/3. An adverse oxidative reaction occurs if the tertiary structure is disrupted where Cu becomes unbound in SOD1/3. Copper then reacts through a Fenton-like reaction, enabling it to generate ^•^OH that is heavily reactive towards biomolecules [129,130]. While the Fenton reaction can induce ROS production from excess Cu, deficiencies in Cu are also attributed to inefficient oxidant removal as it is a cofactor for SOD1/3 [131,132]. Therefore, sufficient levels of Zn are necessary to maintain SOD1/3.

## 5. Antioxidant Potential to Improve Bovine IVP Systems

### 5.1. IVM

Trace minerals are present in IVM media formulations that contain serum. Also, conventional media formulations (e.g., M199) usually contain HTU, cystine, cysteine, and GSH. However, the premise for testing these and other antioxidants during IVM is warranted because these conventional media were not originally designed as oocyte culture media. With that said, some antioxidants have not been examined in detail. Hypotaurine, for example, failed to alter oocyte GSH concentrations when provided during individual oocytes per culture drop during IVM [133], and outcomes like this have undoubtedly reduced enthusiasm for studying this antioxidant during IVM.

Cysteine and cysteamine have been widely researched because of their need for generating GSH within the cell. Supplementation of cysteine during IVM improved oocyte development by increasing the percentage of oocytes that reached metaphase II [92,134,135,136,137]. This supplementation scheme, however, does not always improve bovine blastocyst yield [135,138,139]. Positive effects of cysteine on blastocyst formation are observed more often in stressed environments, such as heat-stressed culture systems [79]. When supplemented during maturation, the cysteine analog, NAC, also improved embryo cleavage and blastocyst development comparable to cysteine [139,140]. Cysteamine appears to be very important for controlling GSH concentrations within the bovine oocyte. Sovernigo et al. (2017) reported that using cysteamine during IVM decreased ROS concentrations and increased GSH concentrations in bovine oocytes [141]. Other studies determined that cysteamine supplementation improved oocyte competency and maturation, increased blastocyst development, improved cryosurvivability, and reduced the adverse effects of prolonged exposure to elevated oxygen tension [9,90,103,135,138,142,143,144,145].

There are only a few studies that explored Se supplementation during IVM. One of these studies combined Se with other trace minerals (Cu, manganese, Zn) [146]. Cleavage rates were not impacted by this cocktail, but improvements in blastocyst formation, cell number, and reduced ROS concentrations were detected on day 8 blastocysts [146]. Another study where Se was the only supplement used had improved blastocyst development with a concentration of 10 ng/mL [147]. When Se was provided at 10 ng/mL during IVM, improvements in cumulus cell viability were observed when compared with lower concentrations, although this treatment group had greater lipid peroxidation compared to lower concentrations. Additionally, oocytes cultured with 10 ng/mL had the greatest intracellular GSH compared to any other treatment [147]. Another study evaluated Se at 10 ng/mL under heat-stressed conditions. Selenium supplementation improved nuclear oocyte maturation compared to the heat-stressed control, with a greater number of oocytes reaching the MII stage [148]. During this study, GPx4, SOD, and CAT had upregulated transcription levels that reduced ROS accumulation [148]. 

The supplementation of Zn during IVM benefits subsequent IVF and in vitro culture (IVC) success. One study found that Zn supplementation at the time of IVM increased subsequent cleavage rates and blastocyst formation [149]. This work also identified increases in blastocyst cell numbers in Zn-treated cumulus oocyte complexes [149]. Interestingly, no differences in intracellular GSH concentrations were detected with Se supplementation, but there was less damage observed in cumulus cells cultured with Zn [149]. A related work identified that Zn supplementation during IVM reduced apoptosis and increased SOD activity in cumulus cells [150,151]. In buffalos, Zn supplementation during IVM supports both oocyte maturation and subsequent blastocyst formation [152].

### 5.2. IVF

Selenium, HTU, and Zn have been examined for their abilities to improve IVF success. The studies focusing on Se and Zn actions were primarily focused on examining how they influence sperm viability and binding to the zona pellucida. Indeed, one study evaluated Cu, Mn, Se, Zn, and a cocktail containing all of these trace minerals [153]. The cocktail mixture did not improve sperm viability, membrane integrity, acrosomal status, or zona pellucida binding [153]. Interestingly, individual supplementation of Cu, Se, or Zn improved sperm binding [153]. While this study did not evaluate pairings of antioxidants, it would be interesting to investigate Cu and Zn supplementation together as they are cofactors for SOD1. Nonetheless, further work seems warranted to explore whether these same outcomes can be obtained in laboratories that may use slightly or substantially different IVF protocols.

Hypotaurine seems like a favorite antioxidant of choice during IVF, and sperm viability is the primary target of its actions. Most laboratories use cryopreserved bovine semen, and adding HTU to the sperm cryopreservation mix improves post-thaw survivability, improves the onset and completion of capacitation, reduces premature chromatin decondensation, limits DNA fragmentation, and reduces the presence of nuclear vacuolization [154]. Supplementation of HTU is also used after sperm is thawed. Its inclusion during IVF improves survivability and reduces chromatin decondensation, DNA fragmentation, and nuclear vacuolization [154]. However, its presence during IVF may produce adverse outcomes. One study supplementing HTU during IVF contained a reduced fertilization rate and increased incidence of polyspermy [133]. Despite this, the promise of HTU as a beneficial antioxidant during IVF is why it is included in the penicillamine, HTU, epinephrine (PHE) cocktail that is used by many laboratories. Penicillamine improves bovine sperm viability and epinephrine facilitates sperm motility [155,156]. Researchers reported mixed results as to whether sperm motility is affected by PHE treatment, but PHE was still able to improve early embryonic development [157,158,159].

Hypotaurine is not the only antioxidant studied for its ability to function during sperm cryopreservation. For example, GSH supplementation improved sperm motility, progressive sperm motility, average velocity, and progressive linear velocity [160]. This GSH supplementation did not, however, improve cleavage or blastocyst rates [160], suggesting that GSH may provide more of a benefit when cryopreserved semen is thawed and inseminated into cattle rather than used for IVF.

Zinc deficiency impairs sperm motility, morphology, and viability in the reproductive system [128,161]. This provides ample justification for supplementing Zn during IVF, although it is unclear whether Zn benefits or hinders IVF success. Stephenson and Brackett reported that supplementing 1 µg/mL of Zn chloride inhibited fertilization success by interfering with calcium oscillations [162]. This may be caused by the interference of Zn efflux that is important for regulating intracellular pH and calcium entry. Also, supplementing Zn during IVF may compromise Zn flow. Zinc efflux during fertilization is referred to as the Zn spark and is required to assist in the resumption of meiosis II and assist in blocking polyspermy [163,164]. However, at least one recent report has not observed a detrimental effect of Zn supplementation, with sperm viability and progressive motility staying elevated at 0.8 μg/mL 6 h after thawing and increased zona pellucida binding [128]. That same study did not observe any positive effects on embryonic development when Zn was supplemented in IVF [128]. Based on these findings, it seems ill-advised to supplement Zn during IVF. There are other examples where antioxidants may produce harmful outcomes during IVF. Cysteamine supplementation, for example, hinders sperm quality, compromises pronuclear formation, and ultimately reduces blastocyst development [165].

### 5.3. IVC

It is interesting that only a limited number of studies have explored antioxidant supplementation during IVC. Antioxidants have been explored during IVC for their ability to limit and/or correct adverse cellular damage caused by heat stress (see [166] for review), but their inclusion is otherwise rare. The rare cysteine and cysteamine supplementation studies have resulted from their inability to manipulate intracellular GSH concentrations during early embryo development [103,139]. At least one study observed reduced blastocyst development after cysteine supplementation [103].

There have been efforts to explore Se and Zn for their abilities to improve IVC success. Studies evaluating Se have used it in conjunction with a serum substitute mix containing insulin/transferrin/Se (ITS). Several studies found that ITS improves blastocyst development [167,168,169,170,171,172,173]. Our laboratory recently identified that Zn supplementation did not impact blastocyst yield but improved blastocyst ICM, trophectoderm, and total cell number compared to control [174]. Further work is needed to verify whether Zn supplementation is useful in different IVC culture conditions, but perhaps more importantly, this work shows that measurements of embryo quality (e.g., cell number and ICM/TE distribution) require more attention as we and others pursue antioxidant activities in bovine IVP systems.

There has been some work involved in supplementing SOD or CAT during IVP, but no benefits of these supplements have been observed [139]. There have, however, been a few studies that have identified positive outcomes from supplementing GSH during IVC. In one study, GSH supplementation did not improve cleavage rates or blastocyst development but improvements in the total cell number and ICM numbers were detected in GSH-supplemented blastocysts [175]. Another study did observe increases in cleavage and blastocyst rates when using a heavy isotope-labeled GSH (GSX) [176]. It is intriguing to consider that antioxidant treatments that only function outside of the cell may be beneficial to in vitro embryo development. More work is needed to explore this in greater detail.

## 6. Conclusions

This review provides ample evidence supporting the contention that oxidative stress hinders bovine IVP outcomes (Figure 4). Stresses likely include temperature fluctuations, exposure to atmospheric O_2_, UV light, pH changes, and cryopreservation-induced cell damage. These and other factors contribute to the generation of ROS. If left uncontrolled, these reactive molecules can damage nucleic acids, lipids, and proteins. If controlled, however, the degree of damage can be limited and quickly corrected. There are several enzyme systems within all cells, including the oocyte and embryo, which mitigate ROS-induced damage.

This review highlighted how five antioxidants may be used within bovine IVP systems to mitigate ROS-induced damage. Studying antioxidant supplementation during IVM has been complicated by the presence of several antioxidants within commonly utilized media formulations. Those media formulations contain trace minerals, again making it difficult to determine if supplementing additional trace minerals is beneficial. However, even with these complications, there is good evidence suggesting that supplementing cysteine, cysteamine, Se, or Zn may have the ability to mature oocytes to be fertilized, cleave, and produce blastocysts. There also is good evidence to support the contention that supplementing Se and/or HTU will improve sperm activity, thereby improving IVF success. It is less clear if Zn supplementation during IVF is beneficial or detrimental. There are numerous studies exploring antioxidant use for mitigating ROS-induced damage during IVC when bovine embryos are exposed to stressors, namely heat stress, but much less work has been completed when examining IVC events in the absence of intense stressors. Sufficient evidence exists to propose that supplementing Se as part of the ITS serum substitute mix during IVC improves blastocyst development. It is unclear, however, specifically which ingredient in the ITS cocktail is responsible for this improved development. There is also some evidence suggesting that Zn supplementation during IVC may improve bovine blastocyst cell numbers and that GSH supplementation may be acting outside of the cell through some undefined means to promote bovine embryo development.

To conclude, it is difficult to make any firm recommendations for supplementing any one or a specific set of antioxidants in bovine IVP systems. The work we have presented here indicates that opportunities exist to promote bovine IVP success using antioxidant supplementation strategies; but, what may work in one bovine IVP system may not work in other systems. Most bovine IVP systems have been tailored to work within specific laboratory conditions, and each laboratory likely will need to undergo empirical testing to identify the specific antioxidant combination that works best for their system.

## Figures and Tables

**Figure 1 animals-14-00330-f001:**
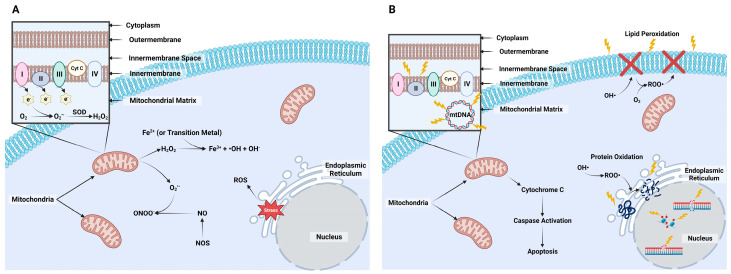
The formation of reduced oxygen species (ROS) and the main types of damage these molecules produce within cells. (**A**) The mitochondria are a predominant source for ROS production when electrons (e^−^) escape from complexes I, II, and III, and will quickly react with oxygen (O_2_) to produce superoxide anion (O_2_^•−^). Superoxide dismutase (SOD) will reduce O_2_^•−^ to hydrogen peroxide (H_2_O_2_), where it can enter the cytoplasm through peroxiporins in the mitochondrial membrane. In the cytoplasm, H_2_O_2_ can further react with transition metals, such as iron (Fe^2+^), to produce the hydroxyl radical (^•^OH) and hydroxide (OH^−^). Nitric oxide synthase (NOS) will generate nitric oxide (NO) in normal amino acid metabolism. However, reactive nitrogen species (RNS) may also be produced from O_2_^•−^ reacting with NO to produce peroxynitrite (ONOO^−^). The endoplasmic reticulum (ER) is also responsible for producing ROS from excess protein synthesis that can lead to misfolded proteins and ER stress. Stressed ER will generate H_2_O_2_. (**B**) Nucleic acids, lipids, and proteins are primary targets for ROS-induced damage. Nucleic acid damage forms lesions and strand breaks in DNA and can alter the methylation profile. Lipid peroxidation of the cell, mitochondria, and nucleus are another target for ROS. Damage occurs when ^•^OH binds to lipids to generate a peroxyl radical that will react with an additional polyunsaturated fatty acid to form hydroperoxide and alkyl radicals (ROO^•^). This feed-forward reaction will compromise membrane fluidity and function. Additionally, mitochondrial membrane permeability may be altered to change the conductance state of the mitochondria. Protein damage is also caused by ROS where ^•^OH binds to generate ROO^•^. Created with Biorender.com (accessed on 12 January 2023).

**Figure 2 animals-14-00330-f002:**
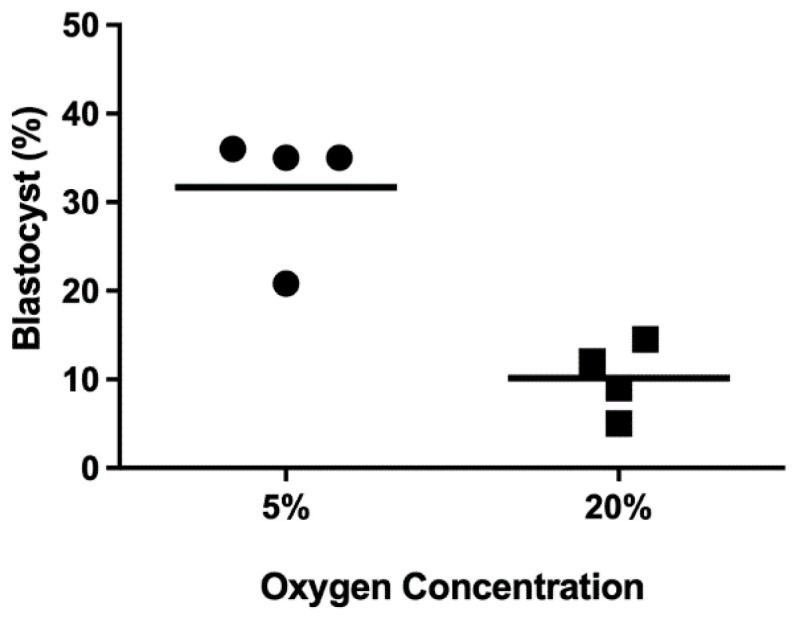
Elevated oxygen concentration during culture has detrimental effects on blastocyst development. These studies evaluated bovine embryonic development cultured in either 5% (individual studies represented as a dot) or 20% oxygen concentration (individual studies represented as a square) for the entirety of the culture period [7,52,54,55].

**Figure 3 animals-14-00330-f003:**
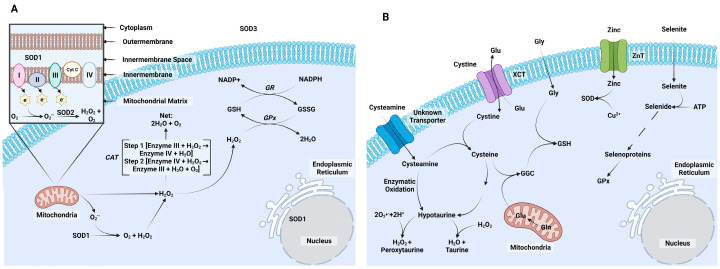
Cellular control of reactive oxygen species (ROS). (**A**) There are three intrinsic mechanisms to reduce ROS. Superoxide dismutase (SOD) is localized in different portions of the cell, with SOD1 (Cu/Zn-SOD) present in the cytoplasm and nucleus, SOD2 (Mn-SOD) present in the mitochondrial matrix, and SOD3 (Extracellular-Cu/Zn-SOD) located extracellularly. These manage ROS by reducing O_2_^•−^ to H_2_O_2_. There are two methods to reduce H_2_O_2_. The first method is by catalase enzyme (CAT), which acts in a two-step reaction to remove two H_2_O_2_ and produce two water and one oxygen. The second method is glutathione and glutathione peroxidase (GPx). The reduced form of glutathione (GSH) is activated by GPx to convert two water molecules from H_2_O_2_. The oxidized form of GSH (GSSG) is replenished by glutathione reductase to remove future ROS. (**B**) Five antioxidant pathways are reviewed herein for their ability to assist in removing ROS. Cysteine is produced in the cytoplasm after cystine is transported through a cystine/glutamate antiporter (XCT). Cysteine contributes to the formation of γ-glutamylcysteine (GGC) after reacting with glutamate (Glu) produced by the mitochondria. Finally, GGC reacts with glycine (Gly) to form GSH. Cysteamine is important as it assists in the conversion of cystine to cysteine. Both cysteine and cysteamine feed into the production of hypotaurine. Hypotaurine reduces H_2_O_2_ to form water and taurine. However, hypotaurine may also produce H_2_O_2_ and peroxytaurine from O_2_^•−^ and hydrogen. Zinc enters the cell through zinc transporters (ZnT) and contributes to the formation of SOD1 (Cu/Zn-SOD). Selenite is commonly included in cultures produced by IVP where it can enter the cell and produce selenide. Selenium is a cofactor for GPx. Created with Biorender.com.

**Figure 4 animals-14-00330-f004:**
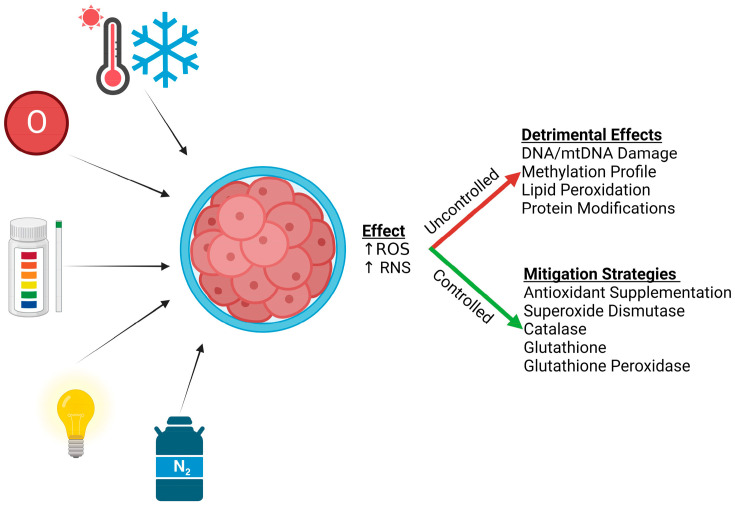
During IVP, bovine oocytes and embryos are assaulted by several external factors that include temperature fluctuations, exposure to atmospheric oxygen, pH changes and light exposure and potentially undergo cryopreservation. These stimuli increase ROS and RNS. When left uncontrolled, substantial damage may occur that includes DNA damage, alterations in methylation profile, lipid peroxidation, and protein modifications. However, if controlled by intrinsic mitigation pathways including SOD, CAT, GSH, and GPx or by antioxidant supplementation, ROS and RNS have a reduced detrimental effect that improves embryonic development. Created with Biorender.com.

## Data Availability

Not applicable.

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
