# Peer review of "An Overview of Reactive Oxygen Species Damage Occurring during In Vitro Bovine Oocyte and Embryo Development and the Efficacy of Antioxidant Use to Limit These Adverse Effects"

_animals, 2024, doi:10.3390/ani14020330_

Round 1

Reviewer 1 Report

Comments and Suggestions for Authors

This review summarizes very important aspects of the antioxidant role in reducing harmful effect of ROS during bovine in vitro oocyte and embryo development. This study expands our insight on the effect of several known antioxidants (cysteine, cysteamine, selenium, hypotaurine, zinc) in oocytes and embryos under normal conditions as well as under stressed conditions (high oxygen content, heat stress, UV-light, cryopreservation etc.).

Interestingly, this review provides evidence for not only positive, but also for harmful influence of several antioxidants on IVF outcomes. This is a strong aspect of this review, because we should keep in mind that positive effect of AOs may greatly depend on their proper concentration and combination with other antioxidants in the AO cocktail.

I suggest authors to expand the review with some observations on the use of glutathione (GSH) to mitigate ROS-induced damages in bovine oocytes or embryos following cryopreservation.

Generally, the review is well-written and properly organized, accompanying with four schemes which simplify the presented data. However, there are some ambiguities and misprints in the text which have to be corrected or clarified:

Lines 17, 19 and everywhere in the text: ROS damage – change for ROS-induced damage.

Line 420: NAC – this abbreviation should be clarified.

Line 420: “…survival from under cryopreservation…” – this phrase has to be edited.

Line 480: “spermcryopreservation” – divide into two words.

Line 498: “exmaples” – to correct the word.

Line 507: “…supplementation studies has likely resulted…” - change for “have…”.

Line 509: “devleopemnt” – to correct the word.

Line 510: “There has been efforts…” - change for “have”.

Lines 521-522: “There has, however, been a few studies…” – change for “have”.

Line 528: “…to explore this is greater detail…” – change for “this in greater detail…”

Comments on the Quality of English Language

My comments and suggestions on the language quality are presented above.

Author Response

Thank you for the excellent review. 

Reviewer 2 Report

Comments and Suggestions for Authors

Overall, a very nice review of antioxidants in in vitro embryo production.

It might be appropriate to include some discussion of the physical environment, particularly oxygen levels and how this frequently differs between the three components of IVP, IVM, IVF and IVC.  It is well accepted that reduced oxygen levels are beneficial for embryo culture, and this is a common practice.  The same cannot be said for IVM and IVF.  Both IVM and IVF are frequently conducted at atmospheric oxygen levels which is very likely to impact ROS and methods to mitigate.

The complete list of compounds used to mitigate the effects of ROS is long but two in particular have received recent attention in this regard, Resveratrol and melatonin.  

Author Response

Thank you for your help with this review.

Reviewer 3 Report

Comments and Suggestions for Authors

In their review authors focused on reactive oxygen/nitrogen species and their influence on in vitro bovine oocyte and embryo development.

In the first part of the review, authors provide extensive, in-depth description of reactive oxygen (ROS) and nitrogen (RNS) species and their mechanisms of action. In the second part of the review, authors discuss selected antioxidants and their usage during in vitro bovine oocyte and embryo culture.

The review is very well written, illustrations are sufficient and all important problems are discussed adequately.

First part is very detailed (may be even too much compared to the second part), but this should be OK. I have 2 comments:

1)      ROS are known to induce intracellular signaling (and influence many processes in non-detrimental way) when present at physiological levels. However, this is not at all mentioned in the review. I would suggest to include a section dealing with physiological function as well to illustrate how complex the ROS system is. Also, role of ROS/RNS signaling during oocyte/embryo maturation could be discussed, if any.

2)      as this part is really dominating the review, authors should consider to change/adjust review’s title accordingly

Comments to the second part:

1)      selection of antioxidants mentioned in the section 4 is not much clear to me. How exactly were these antioxidants selected? Based on which criteria? I guess that many more molecules (“endogenous” and even “exogenous”) have been tested during in vitro maturation.

2)      please explain abbreviations IVM, IVF, IVC, RNS in the main text when used for the first time

3)      check sentences at lines 229-231. What does it exactly mean “fluorescent lightning”? Do authors mean UV part of daylight? Please explain.

4)      please, check (and rephrase) lines 292-297. Sentences seem to be duplicated and the meaning is not exactly clear

5)      sections 4.1 and 4.2 should be carefully checked and potentially rephrased

Comments on the Quality of English Language

English is OK. I pointed out just several potential typos:

-line 14: … and embryo to reactive …

-line 16: developmental

-line 29: … ROS produce …

-line 30: … alter DNA methylation, increase lipid membrane damage, and modify protein …

-line 137: More important for …

-line 186: … the caspase pathway.

-line 218: … vary between 2-9% …

-line 229: -line 278: … first is SOD.

-line -285: …convert it to …

-line 321: rephrase “Once time”

-line 323: … GSH level is at …

-line 369: … selenocysteine [109-111]).

-line 480: …sperm cryopreservation.

Author Response

Thank you for taking the time to review our manuscript.
